# Neurobiology of Pathogen Avoidance and Mate Choice: Current and Future Directions

**DOI:** 10.3390/ani14020296

**Published:** 2024-01-17

**Authors:** Dante Cantini, Elena Choleris, Martin Kavaliers

**Affiliations:** 1Department of Psychology, College of Social and Applied Human Sciences, University of Guelph, Guelph, ON N1G 2W1, Canada; dcantini@uoguelph.ca; 2Department of Psychology, Western University, London, ON N6A 3K7, Canada

**Keywords:** social brain, disgust, parasites, neuromodulation, mate preference

## Abstract

**Simple Summary:**

The risk of parasitic infection has a major influence on animal behaviour. Organisms must adjust their behaviour to avoid various modes of parasitic infection and pathogen acquisition. Social species are at an increased risk of parasitic transmission as they spend more time in the proximity of others that may carry parasites. The detection of parasitic risk is also critical in mate assessment and choice. Perceptual systems and behavioural responses have evolved to detect individuals who are parasitized and pose the risk of parasitic transmission. This includes the integration of inputs from various sensory modalities (e.g., olfaction), brain regions and networks, and neuromodulatory systems. Understanding the neurobiological systems involved in detecting the parasite infection risk and the expression of disgust will allow us to better understand the evolution and regulation of pathogen avoidance and mate choice.

**Abstract:**

Animals are under constant threat of parasitic infection. This has influenced the evolution of social behaviour and has strong implications for sexual selection and mate choice. Animals assess the infection status of conspecifics based on various sensory cues, with odours/chemical signals and the olfactory system playing a particularly important role. The detection of chemical cues and subsequent processing of the infection threat that they pose facilitates the expression of disgust, fear, anxiety, and adaptive avoidance behaviours. In this selective review, drawing primarily from rodent studies, the neurobiological mechanisms underlying the detection and assessment of infection status and their relations to mate choice are briefly considered. Firstly, we offer a brief overview of the aspects of mate choice that are relevant to pathogen avoidance. Then, we specifically focus on the olfactory detection of and responses to conspecific cues of parasitic infection, followed by a brief overview of the neurobiological systems underlying the elicitation of disgust and the expression of avoidance of the pathogen threat. Throughout, we focus on current findings and provide suggestions for future directions and research.

## 1. Introduction

For many species, living in social groups provides evolutionary benefits, such as protection from predators, sharing of information and resources, and increased potential mating opportunities [1,2,3,4,5]. Living in large social groups also has costs and puts individuals at an increased risk of infection [6,7,8]. There is evidence that a larger social group size positively correlates with an increased individual risk of disease and parasitic infection, largely attributed to an increased contact rate among individuals [8,9,10,11] (for further discussions, see Altizer et al. (2003) [9]). Contrary to the view of parasitic risk being a cost of social groups, it is speculated that group living may confer anti-parasitic benefits via resistance to parasites (i.e., social immunity) and enhanced tolerance of parasitic infection (i.e., social interactions that improve physical condition via increased resource acquisition or social support) [10,12]. Nonetheless, the risk of parasitic infection via conspecific transmission is an integral aspect of group living that shapes the evolution of sociality.

In the wild, individuals of most social and non-social species harbour varying levels of parasites and pathogens, including harmful bacteria and viruses. For example, members of natural mouse populations frequently carry some form of parasite, such as skin mites and various nematode species [13,14]. Gregory et al. (1992) [14] proposed that approximately 80% of all wild wood mice are likely to be infected with a parasitic load of 10 to 25 helminth parasites. In natural environments, as opposed to most sterile laboratory environments, it is unlikely to find organisms that are fully parasite-free [15]. Here, we will refer to parasites broadly to encapsulate any relationship in which one organism (the parasite) benefits by living in or on the host organism at the cost of the host [16].

The prevalence of parasites in wild populations puts evolutionary pressure on individuals to assess and avoid infection risk. Detection and avoidance of parasites is beneficial for individuals at both the proximate level (avoiding direct transmission of parasites and disease to oneself) and the ultimate level (selection of genes that will improve the disease resistance of one’s offspring). To avoid infection, animals must modulate their behaviour and interactions with conspecifics based on social information that conveys the health status of others. 

Curtis and Biran (2001) [17] proposed disgust as an evolutionary adaptation that biases behaviour toward the avoidance of parasitic infection, contaminants, and toxin risk. Disgust is proposed as a fundamental affective/emotional state that underlies and facilitates the behavioural avoidance of individuals that are perceived on the basis of various sensory cues that represent a potential risk of pathogen exposure and infection [18,19,20]. In addition, information regarding infection or contagion can elicit other forms of affect, such as fear and anxiety [19,21].

It has been further proposed that individuals face a ‘landscape of disgust’, in which animals must detect ‘mountains’ of high infection risk and ‘valleys’ of low infection risk [22]. The ecological and evolutionary consequences of an individual’s responses have been framed in terms of the ecology of disgust [19]. This is particularly pertinent for social behaviours, including mate choice and partner assessment. Individuals who are better at processing sensory information, which leads to appropriate disgust responses and elicitation of avoidance, are less likely to come into contact with parasites, which reduces their probability of infection. 

In this selective review, we will consider aspects of the neurobiological behavioural mechanisms underlying parasite detection and avoidance and their implications for mate choice in rodents. We will initially provide a brief overview of aspects of (i) mate choice relevant to pathogen avoidance. Then, we will specifically focus on (ii) the olfactory detection of and response to conspecific cues of parasitic infection and immune status. Finally, (iii) the neurobiology underlying the elicitation of disgust and expression of avoidance and aversive responses to pathogen threat (see Figure 1 for a visual representation of the topics covered in this review). Focusing on select current findings, we will provide suggestions for future research.

## 2. Aspects of Mate Choice Relevant to Pathogen Avoidance

### 2.1. Mate Choice

Mice and many other rodents have a polygamous mating system, meaning that both males and females will mate with multiple different partners throughout their reproductive lifespan [23]. A polygamous mating system makes these rodents particularly susceptible to infection by contagious parasites due to the increased frequency of physical proximity and sexual contact with multiple individuals [9,19]. Appropriate assessment and detection of parasitic infection in conspecifics is especially pertinent during mate choice, as there can be a direct risk to the individual through sexually transmitted parasites, such as the protozoan parasite, *Toxoplasma gondii* [24]. The risk of infection creates a strong selective pressure on the ability to detect actually and potentially parasitized individuals. Many vertebrates, including rodents, use odour cues to detect parasitic infection, and these odours also function as olfactory phenotypes that inform mate choice e.g., [25,26,27,28,29]. Phenotypic secondary sexual characteristics that are generally used to assess a potential mate’s viability can also serve as an indication of the individual’s parasitic load; this has been termed the ‘contagion indicator’ hypothesis [30]. 

The strong selective pressure to appropriately assess a potential mate’s infection status has both direct and indirect evolutionary benefits for the individual [24,30,31]. Loehle (1995) [32] theorized that females are more risk-averse to disease threats. As a result, female aversion to parasitized males has likely evolved to both avoid direct transmission of contagious pathogens during mating and select for heritable parasite resistance [24,30]. Thus, the selection of non-parasitized mates is beneficial at both the proximate level (avoiding direct transmission of parasites to oneself) and the ultimate level (selection of genes that will improve the disease resistance of one’s offspring). Evidence to support the ultimate-level benefits is provided by the finding that female mice that chose males with ‘maximal’ immune function produced offspring with more robust immune systems [33]. As such, the ability to recognize and avoid infected, or potentially infected, individuals is a key component of mate choice.

In rodents, and most other mammals, sexual behaviour can be divided into two phases: the appetitive (precopulatory) and consummatory (copulatory) phases [34]. The appetitive phase involves the motivation to approach and engage in an assessment of a potential mate [35]. This has also been considered in terms of incentive salience and the rewarding value of the potential mate [36,37,38]. Mate choice, which encompasses the appetitive phase, can be divided into two categories: preference (the preferential order of attractiveness in which an individual ranks potential mates) and choosiness (the responsiveness to potential mates and the amount of effort expended to assess the mate and make the choice) [19,39]. In theory, if all potential mates are available, the chosen mate should closely reflect the peak mate preference. Within the context of parasitic infection, if an uninfected versus an infected mate is available to a healthy individual, the uninfected mate is generally preferred [40]. However, preference and mate choice are not always mutually exclusive. It should be noted that some researchers believe that choice can only be said to have occurred after mating. Preferences are inferred from choices that are made using a proxy (e.g., odours) for mate preferences. For a detailed consideration of the design of mate choice experiments and the use/measure of choices and preferences, see Dougherty (2020) [41] and Dougherty (2023) [42] and the tables and references therein. 

Patterns of mate choice can be altered by changing the costs of choosiness without altering the mate preference (i.e., without changing who would be the most preferable mate). As indicated, in the wild, as opposed to laboratory conditions, there may not be a direct choice but rather potential sequential encounters or odour exposures. This constraint on choosiness results in uncertainty regarding the condition and infection status of subsequent potential mates and their reward value. Indeed, the results of studies with rodents in semi-natural environments have suggested that female copulation and impregnation do not necessarily correspond to perceived female mate preference [43]. In a semi-naturalistic study on rats, Chu and Ågmo (2014) [43] demonstrated that when the female preference was defined by the frequency of female sniffing of the male (a behaviour typically associated with preference in most classical laboratory studies), there was no difference in sexual behaviour among the preferred and non-preferred males, and thus female preference within these parameters did not impact male reproductive success. This suggests that in naturalistic conditions, a preferred male may not produce more offspring than a non-preferred male. 

Therefore, in nature, not all individuals have the choice to mate with the preferred individual, and thus mate choice is not always representative of mate preference. As a result, female preference for non-parasitized individuals is not always evident and is influenced by other variables. Female mice that are pre-exposed to the odour of parasitized males do not show an initial preference for non-parasitized male odour [26]. Additionally, it was found that a parasitized male odour paired with the odour of an estrous female attenuated the aversive response and resulted in increased female preference for the parasitized male odour [44]. Individuals pay attention to the social and mate decisions of conspecifics and use them to determine their own choices, displaying mate choice copying [45,46]. These findings suggest that the preference for healthy mates can be influenced by the social context and indirect social information that results in ‘mate choice copying’, negating the notion that initial preference will always be directed to the non-parasitized potential mate [46,47]. Whether or not mate choice copying is utilized to avoid potentially infected individuals remains to be determined. However, there is evidence that female mice will avoid the odours of uninfected males that have been associated with those of infected males [48].

The influence of the parasitic infection risk on sexual behaviour and mate choice heavily influences sexual selection [40,49]. Despite some notable exceptions, animals generally demonstrate preference and choice for parasite-free mates to increase both direct and indirect fitness [50,51,52,53,54]. To adaptively reproduce, animals must engage in appropriate behavioural responses when faced with potential infection risks. These behavioural responses not only influence the transfer of genes in a population but also serve to protect the individual against parasitism as well as mitigate the pervasiveness of parasites at the population level. As we briefly discuss in the following sections, the affective disgust response is integral to the behavioural responses that influence mate choice regarding the parasitic infection risk.

### 2.2. Approach and Avoidance 

Loehle (1995) [32] postulated that organisms may employ ‘social barriers’ to mitigate the infection risk, suggesting that various social behaviours (e.g., general social avoidance and avoidance of mating with infected individuals) may effectively reduce pathogen transmission. Hamilton and Zuk (1982) [55] were among the first to directly hypothesize that animals benefit from inspecting a conspecific’s urinary and fecal odours to select a parasite-free or resistant mate. The assessment of infection status facilitates the expression of the appropriate adaptive social response directed toward the conspecific. The process underlying the social assessment of conspecifics involves social cognition, which includes the acquisition of social information that is used to assess the status of other individuals [19,56]. Social cognition allows individuals to rapidly mediate their behaviour toward conspecifics, including approach and avoidance, in a manner that is adaptive within dynamic social contexts [19].

Approach can be defined as the direction of behaviour toward positive stimuli (higher incentive salience and reward), while avoidance can be defined as directing behaviour away from negative stimuli [57]. Approach and avoidance behaviour can be directed toward objects, individuals, and situations. In terms of the infection risk, avoidance can be defined as any host defence that reduces the rate of contact between a potential host and parasite [16]. Pathogen avoidance behaviour can be undertaken through: (i) directly avoiding or removing visible parasites; or (ii) indirectly avoiding individuals or altering interactions with individuals that may be a potential source of parasites and infection [19,56,58]. Once a pathogen threat has been detected, social avoidance serves as the first behavioural defence against infection that an individual mounts and is likely the most effective action to protect against infection [16]. This has been termed the behavioural immune system in human studies [59]. Pathogen avoidance is, however, context-dependent and needs to be balanced against other behaviours that are essential for survival and reproduction. Context-related factors (e.g., food availability, predator exposure, etc.) need to be incorporated into future laboratory studies.

Behavioural avoidance of parasitized and immuno-activated conspecifics has been documented in various rodent species. The odours of healthy conspecifics have been shown to elicit approach behaviours, while the odours of individuals with infection or immune activation are shown to induce avoidance [19,60,61]. For example, treatment of female mice with the endotoxin lipopolysaccharide (LPS) resulted in various behavioural avoidance responses, such as increased inter-individual distance, decreased rates of physical contact, and the modality of social exploration (i.e., decreased anogenital sniffing) [62]. Similarly, male rats demonstrated significant suppression of social interaction directed toward LPS-treated male conspecifics [63]. Additionally, female prairie voles spent more time in close proximity to a healthy male in comparison to an LPS-treated male [64].

Regarding zoonotic parasites specifically, multiple studies have demonstrated that rodents tend to also avoid conspecifics carrying nematode parasites (i.e., helminths) and ectoparasites (i.e., lice and ticks). Female mice avoid the urinary odours of males infected with the louse *Polyplax serrata* [54]. Edwards (1988) [65] found that healthy male mice engaged in increased olfactory investigation of conspecific males infected with *Trichinella spiralis* but engaged in less physical contact, suggesting the processing of olfactory cues associated with infection is important in the elicitation of avoidance behaviour. Female rats, mice, and meadow voles have also all been shown to similarly discriminate against males infected with various species of parasitic nematodes [28,66,67]. 

## 3. Olfactory Mediated Detection of and Response to Infection Status

Rodents rely heavily on olfaction to provide social information about the status and condition of conspecifics, including that of potential mates and social partners. Chemosignals are informative of an individual’s identity and condition, including infection status [53,68,69,70,71]. The rodent olfactory system is subdivided into two primary components, the main olfactory system (consisting of the main olfactory epithelium (MOE)) and the vomeronasal system (VNO) [72,73,74]. The MOE projects to the main olfactory bulb, while the VNO projects to the accessory olfactory bulb [74]. Rodents emit various odours consisting of volatile (detected directly through the MOE) and non-volatile (detected through the VNO) chemical constituents that serve as an indication of a conspecific’s physiological state [23,75]. In mice, pheromone chemical cues are typically present in urine and convey information about health status [76,77]. However, the roles of other odour sources, particularly that of the preputial gland in males, need to be further examined (for a discussion of odour sources and sexual attraction, see Le Moëne and Ågmo (2018) [78]).

Non-volatile olfactory cues in the bodily products of mice trigger neural activity in the VNO, which is specifically involved in the expression of various innate behaviours [73,79]. Boillat et al. (2015) [73] demonstrated that mice with impaired VNO function did not demonstrate the typical preference toward healthy conspecific versus conspecifics treated with LPS. The researchers verified that the aversive infection cues detected by the VNO were present in bedding and urine specifically, as mice with intact VNO function preferentially interacted with bedding materials from healthy conspecific over those of LPS-treated individuals. It should be noted here that bedding contains odour cues from a range of bodily sources. However, the results of cFos imaging found that exposure to LPS-infected urine resulted in higher neural activity in the accessory olfactory bulb, the region to which the VNO projects. 

Although the MOE is not considered to be primarily involved in the detection of pheromones, there is evidence that pheromone-based behaviour is mediated by both the MOE and the main olfactory bulb [74,80,81]. The main olfactory bulb in mice has small populations of mitral cells that respond to volatile urine compounds. Lin et al. (2005) [81] demonstrated that in female mice, subsets of these olfactory mitral cells responded exclusively to male urine due to a novel male-specific pheromone, (methylthio)methanethiol. When added to urine, synthetic (methylthio)methanethiol enhanced female attraction to male urine, suggesting that it may be a volatile compound processed by the MOE and main olfactory bulb that serves in the female assessment of males. Whether or not these constituents are involved in mate choice remains to be determined.

Evidence suggests that urinary odours are involved in conveying information influencing mate choice in rodents. For example, the results of studies have shown that female laboratory mice can discriminate the urinary odours of male mice infected with the helminth *Heligmosomoides polygyrus* and display a preference for the urinary odours of uninfected males [29,82]. Females may be responding to subtle differences in male urinary odours, likely associated with polymorphic gene complexes: the major histocompatibility complex (MHC) and the major urinary proteins (MUP) [68,83]. These gene complexes convey an array of information about sex, kinship, dominance, age, and health status [23]. Olfactory processing of odours containing MHC in particular serves as an indication of genotype and physiological quality and is used to assess immune function [84,85,86]. MUPs have considerable polymorphic expression between individuals and serve mainly as markers of genetic identity (i.e., kinship, genetic heterozygosity), and they are also implicated in the assessment of immune status [76,87]. Only these urinary compounds were sufficient for the detection of conspecific infection status, as feces and bile acid from mice infected with the endotoxin LPS did not induce avoidance in conspecifics [88]. The detection and processing of volatile urinary components allows individuals to identify the producers of infected urine from a distance and avoid close or direct contact with any potential pathogen products.

Parasitic infection has been shown to affect MUP expression in mice. Li et al. (2023) [76] demonstrated that the urine of male mice infected with *Cryptosporidium parvum* displayed lower attractiveness to females, and they additionally showed that the protein concentration of MUP in the urine was downregulated in parasitized males. In addition to this, the contents of different pheromones were downregulated in the preputial gland of these males, suggesting that parasitism alters the concentration of male pheromones. An additional study by Li et al. (2023) [77] demonstrated that the parasite *Trichinella spiralis* had similar effects on male MUP expression and decreased sperm quality. 

This short discussion reveals that a range of volatile and non-volatile pheromone attractants are produced by male and female rodents. These chemical factors are genetically controlled and likely influenced by the social and non-social environmental context (e.g., the presence of conspecifics of varying status (dominance) and physiological condition). There is evidence that mouse urinary volatiles vary according to the pathogen responsible for producing inflammation [89]. As well, the extent of avoidance has been linked to the degree of infection [90]. How infection affects these chemical signals, both in terms of quality and quantity, remains to be resolved. Comparative transcriptomics analysis examining changes in gene expression in relation to odorants in infected and non-infected individuals is required to complement the analysis of the constituents. 

## 4. Neurobiological Activation of Affective and Cognitive Responses to the Pathogen Threat

In recent years, the neurobiological mechanisms underlying the assessment of the infection status of conspecifics have gained significant interest [91]. Here, we will discuss this research and briefly consider the brain regions, hormones, and circuits involved in the expression of disgust and pathogen avoidance in rodents. 

### 4.1. Brain Regions

The processing of pathogen detection and the initiation of appropriate responses involve various brain regions, including the evolutionary conserved social decision-making network. This includes areas such as the social brain and the mesocorticolimbic reward network, which are composed of areas such as the medial amygdala, insular cortex, and related regions [92,93,94]. These are briefly considered below (see Figure 2 for a diagram of these regions and a simplified depiction of their projections relevant to pathogen avoidance). 

#### 4.1.1. Medial Amygdala

The medial amygdala (MeA) is a critical region for the decoding of social stimuli and processing of appropriate behavioural responses. In rodents, both the main and accessory olfactory bulbs have projections directly to the MeA [95,96,97]. The majority of the olfactory projections to the MeA come from the accessory olfactory bulb (and indirectly from the VNO), suggesting that the MeA is involved in the processing of non-volatile chemical sensory information primarily detected by the accessory olfactory bulb [98,99]. In turn, the MeA is functionally connected to a broad network of limbic regions and the hypothalamus, suggesting that the MeA is involved in the transformation of sensory information into specific behavioural responses [98,100,101]. 

The MeA is differentially implicated in both appetitive and aversive/avoidance behaviour, with it and its projections mediating appetitive responses to reproductive stimuli, as well as the processing and avoidance of aversive odours [101,102,103]. The integration of olfactory signal and social stimuli processing in the MeA likely makes the region integral in both the detection of, and the behavioural response to, parasitic risk. 

In male rats, odour from conspecifics treated with LPS has been shown to induce avoidance response in healthy rats. Arakawa et al. (2010) [104] found that the MeA and multiple brain regions that it projects to were activated (including the bed nucleus of the stria terminalis (BNST) and the paraventricular nucleus of the hypothalamus (PVN)) in rats exposed to the odour of a sick LPS-treated conspecific, exemplifying the role of the MeA as a hub of social information processing. In this regard, whether the responses were elicited by exposure to sickness, which elicits a constellation of responses, or to LPS treatment per se remains to be determined. 

Kwon et al. (2021) [105] identified the posteromedial nucleus of the cortical amygdala (COApm) as a region involved in the suppression of mating behaviour in male mice when exposed to an LPS-treated female. Using functional imaging, the COApm was shown to project to populations of glutamatergic neurons in the MeA. Interestingly, Kwon et al. (2021) [105] also demonstrated that the thyrotropin-releasing hormone (TRH) and its receptor within the MeA are implicated in the suppression of mating with LPS-treated females. Male mice with genetically ablated TRH receptors engaged in mating with unhealthy individuals, suggesting that these receptors in the MeA are involved in avoiding interactions and mating with conspecifics displaying cues of enhanced immune activity. 

A recent study investigating the Gai2 vomeronasal system in mouse sickness avoidance behaviour found that the Gai2 pathway is required for the detection and avoidance of LPS-treated conspecific urine [87]. Gai2-dependent neuronal activity occurred in the MeApv when exposed to the urine of LPS-treated conspecifics, suggesting that the region is involved in the detection and avoidance of sick conspecifics. Interestingly, the neuronal activation was not as strong in mice exposed to LPS-treated urine compared to the neuronal activation of control mice exposed to non-infected urine. This may be indicative of the region’s role in social processing and motivation, with the lower MeApv activity in response to LPS-urine potentially contributing to the lack of approach behaviour. 

Together, these findings indicate that the MeA serves as one of the first processing points of conspecific chemosignals, facilitating the detection of pathogenic threats and eliciting subsequent behavioural responses, such as approach and mate choice. As such, the MeA is crucial to the detection of social signals that are indicative of pathogenic threats and likely plays a key role in triggering behavioural avoidance and disgust responses through its projections to other brain regions. Which specific brain regions, and particularly how these may be integrated to facilitate behavioural responses, remains to be determined. 

#### 4.1.2. Insular Cortex

The insular cortex is a region heavily implicated in the disgust response and the mediation of various aversive/avoidance behaviours through connections with both social brain regions and reward regions [91,106,107,108]. The insular cortex can be divided into two distinct regions: the posterior insular cortex and the anterior insular cortex. The posterior insular cortex projects to the sensory and limbic brain regions, while the anterior insular cortex mainly projects to the social brain network, including the medial prefrontal cortex, the nucleus accumbens, and the amygdala [108,109]. 

The insular cortex is directly involved in pathogen avoidance, with inactivation of the posterior insular cortex eliminating the avoidance response to conspecifics treated with a viral mimetic in both female and male rats [110]. A recent study by Min et al. (2023) [109] demonstrated that lesions on the anterior insular cortex of male mice resulted in an inability to distinguish between a cage mate and a novel mouse. Due to the role of the anterior insular cortex in processing information leading to social recognition, this region may also be involved in the processing of infection cues and the recognition, and avoidance, of parasitized individuals. 

A study investigating the correlation between the stereotyped facial expression and the affective state in mice found that exposure to a toxin (lithium chloride) elicited facial expressions that were associated with disgust and corresponded to increased activity in the anterior insular cortex [111]. It should be noted that to date, there are no reports that exposure to the odours of infected individuals elicits disgust-like facial expressions in rodents. More detailed machine-learning assessments of facial responses to infection cues are needed. 

In mice, the insular cortex also receives monosynaptic input from the MeA, with this anatomical connection being considered to be important in valence processing and emotional regulation [109,112,113]. Additionally, the expression of a disgust-like response to toxins in mice involves neural pathways from the anterior insula to the medial and basolateral amygdala [114]. These neural projections between the insular cortex and the MeA denote a functional link between the two regions in the processing of aversive social stimuli and disgust-like responses. 

#### 4.1.3. Habenula

The habenula is a highly conserved limbic brain region implicated in divergent motivational states and cognition [115]. Subnuclei of the habenula are thought to be involved in the processing of avoidance responses to various stimuli. The lateral habenula is central to the brain’s reward system and inhibits dopamine release in the midbrain to signal negative valence [116]. Studies using fibre-photometry found that the lateral habenula of mice shows increased neuronal activity when the mouse was presented with a bitter taste, pain, and aggression from a conspecific [117]. In addition to this, the lateral habenula is also involved in the regulation of social behaviour, with chemogenetic activation of the region resulting in a reduction in social behaviour and social interactions [93,118]. Wang et al. (2017) [117] suggested that aversion processing in the lateral habenula is highly influenced by learning, as a prolonged positive reward inhibited lateral habenula activity in mice that were conditioned to an aversive stimulus. This suggests that lateral habenula activity in response to, and processing of, aversive stimuli is experience dependent. 

Weiss et al. (2023) [87] further implicated the lateral habenula in the detection and avoidance of the urine of LPS-treated conspecifics. Exposure to LPS-treated conspecific urine resulted in lateral habenula activation four times stronger than the activity seen during exposure to control urine. Although the habenula is often described as the region associated with hedonia and anhedonia, more consideration should be focused on this region when studying disgust and aversion to pathogens, as positive and negative affect likely heavily influence adaptive responses to stimuli that evoke disgust. 

#### 4.1.4. Nucleus Accumbens

Recent research has explored the role of the nucleus accumbens (NAc) in the avoidance of aversive stimuli. The NAc is integral in the detection of aversive stimuli and translating motivation into action. The NAc is a convergence point of both reward and aversion circuitry, receiving projections from the ventral tegmental area, the medial prefrontal cortex, and the basolateral amygdala [119,120,121,122]. Specifically, dopamine signalling from the ventral tegmental area has been linked to reward and aversion processing in the NAc, while glutamatergic signalling from the thalamic paraventricular nucleus to the NAc is implicated in aversion [123,124]. He et al. (2023) [119] found that Tac1 neurons in the medial shell of the NAc are specifically implicated in the mediation of the avoidance response to aversive olfactory stimuli (formaldehyde) by receiving excitatory glutamatergic input from the medial prefrontal cortex and sending inhibitory signals to the lateral hypothalamic area. 

The NAc is also involved in social reward, sexual behaviour, and mating [125,126,127]. In both male and female mice, approach and investigation of an opposite sex conspecific result in dopamine activation in the NAc, while dopamine signalling in the NAc is sexually dimorphic during sexual behaviours [128]. Additionally, the role of the neuropeptide oxytocin (which is highly implicated in social behaviour) in the NAc has been shown to regulate social preference and other aspects of social behaviour related to sexual behaviour [129,130,131]. 

Given the role of the NAc in both sexual behaviour and approach/avoidance, it is likely that it is an important region in mate assessment and avoidance of parasitic risk. The NAc and other regions of the mesolimbic reward network are associated with the determination of the incentive salience and reward value of a potential mate [38,132,133]. As such, how the cues of infected individuals influence the functioning of the reward network components (specifically dopamine levels) and subsequent approach/avoidance regarding mate choice remains to be determined. 

#### 4.1.5. Concluding Statement on Brain Regions

This short discussion summarizes the brain regions and their interconnections (circuits) shown to be involved in the mediation of responses to infection threats in rodents. Whether these regions and circuits are similarly involved in the initiation and maintenance of avoidance responses remains to be determined. In addition, a range of other brain regions, including those in the social brain network (e.g., medial prefrontal cortex, basal ganglia, ventral tegmental area), are implicated in the processing of defensive responses to a variety of threats, likely including those to the pathogen threat (for a discussion, see Tseng et al., 2023 [134]). Notably, the prefrontal cortex has been recently shown to encode both general and specific threat representations [135]. In this regard, studies with humans showed that pictures of sick LPS-treated individuals led to decreased activity in the ventromedial prefrontal cortex and the initiation of avoidance [136]. 

#### 4.1.6. Immune–Brain Interactions

The roles of peripheral systems (e.g., immune systems) and their interactions with central systems in the modulation of the behavioural avoidance of the pathogen threat need consideration. Infection- and disease-associated contexts have been shown to trigger immune responses [63,137,138]. It is likely that social cues of infection can alter immune responses in uninfected individuals. Indeed, recent work has suggested a co-evolution of social behaviour and immune responses. A zinc finger transcription factor, ZFP189, was shown to modulate social behaviour by controlling transposable elements and immune responses in the prefrontal cortex of mice [139]. This was based on earlier work suggesting a co-evolutionary link between social behaviour and anti-pathogen immune responses driven by interferon-γ signalling [140]. ZFP182 upregulated immune genes that are involved in anti-pathogen responses, while also promoting social behaviour. Moreover, these responses were suggested to be responsive to environmental stressors, including possibly that of the parasite/infection threat. It was further speculated that other transcription factors may regulate social behaviour and immune responses. These findings provide an exciting new direction for the elucidation of the neurobiological regulation of pathogen avoidance. 

#### 4.1.7. Microbiome–Immune–Brain Interactions 

Animals have diverse populations of bacteria and viruses both internally and externally [141,142,143,144]. There is now substantial evidence that the gastrointestinal microbiota (microbiome) can signal to the brain (microbiota–gut–brain axis) through a range of pathways, including immune activation; production of microbial metabolites and peptides; alterations in neurotransmitters and hormonal levels; neuromodulators and activation of the vagus nerve [145]. These actions can influence various brain regions, including those of the social brain, with marked implications for pathogen avoidance and mate choice. 

The microbiome plays a role in influencing the immune response and, subsequently, social behaviour and pathogen avoidance. The microbiome can influence social behaviour through the production of, and effect on, chemosignals [141,146,147]. Microbial metabolism can release volatile compounds that are detectable by conspecifics and may influence olfactory social signalling [148]. Studies in fruit flies (*Drosophila melanogaster*) have shown that the gut microbiota composition mediates olfactory-based mate preference [149]. More pertinent to this review, Li et al. (2013) [150] demonstrated that trimethylamine (TMA), which is a volatile compound involved in the signalling of species identity, is influenced by the microbiota in mice. TMA is produced by gut bacteria during choline metabolism and is converted into an ‘odorless’ compound by enzymes in the liver. However, these liver enzymes are downregulated in sexually mature male mice, allowing TMA and its odour to be detectable in the urine. When male mice were administered antibiotics, they produced significant quantities of TMA, suggesting the direct implication of the microbiome in producing this chemosignal.

Additional studies have shown that the sexual attractiveness of female mice may be impacted by dysbiosis of the gut microbiome [151]. Yi and Cha (2022) [151] found that male mice showed more interaction with control females in a three-chamber social test in comparison to females that had a gut microbiome manipulated by antibiotic treatment. The authors suggested that the decrease in sexual attractiveness in antibiotic-treated females may be due to alterations to the immune system [151]. It is also possible that dysbiosis of females may be influencing the chemosignals through hormonal effects as well. Some research has coined the term ‘Estrogen–Gut Microbiome Axis’, highlighting the influence that the microbiome has on the estrogenic system [152,153]. Multiple studies have shown that alterations to the circulating estrogen and androgen levels impact gut microbiota composition [154,155,156,157,158,159]. Given that estrogens are heavily implicated in olfactory and pheromonal cues, it is likely that reduced female attraction caused by dysbiosis may be a result of altered estrogen levels. Altered estrogen levels may influence the production of other pheromones and ultimately influence the attractiveness of the female [160,161].

Additionally, some interesting research has directly investigated the influence of parasitism on the microbiome. These interactions are complex and are mediated by the environment, genetics, and the infection burden [162]. Zhao et al. (2019) [163] demonstrated that female mice infected with the helminth parasite *Schistosoma japonicum* had a significant influence on the gut microbiome composition, likely caused by the parasitic egg granulomas in the intestinal tissue. Preliminary data show that the gut microbiome composition is affected by tick parasitism in mice [164]. A number of additional studies in rodents have shown an interaction between parasitic infection and the host microbiome [163,165,166,167]. 

To date, much of the work on the effect of the microbiome composition and how it influences mate choice has been conducted in non-rodent species [149,168,169]. This leaves an exciting avenue for rodent researchers to investigate and establish how the microbiome composition (including both bacterial and viral elements), and the impact of parasitism on this composition, may influence mate choice and social behaviour. Integral to this is also determining how the microbiome influences the expression of affective states such as disgust, fear, and anxiety. 

### 4.2. Neuropeptides and Hormones Involved in Pathogen Avoidance 

Many of the brain regions, and likely the microbiome components, considered above involve the actions of various neuromodulatory systems: various neuropeptide systems, neurotransmitter systems, sex steroid hormones and other steroids. These are briefly considered below. 

#### 4.2.1. Oxytocin

The neuropeptide oxytocin is implicated in various aspects of social behaviour, such as the processing of social information and social memory in mice and various other rodent species [170,171]. In addition to mediating social recognition and peer affiliation, oxytocin is involved in the olfactory-mediated recognition, and avoidance of, infected individuals. In rodents, oxytocin receptors modulate the ‘social salience network’, a collection of interconnected brain nuclei involved in the processing and encoding of social and sensory cues and involved in the processing of cues associated with the expression of disgust [171,172,173,174]. 

Female mice with both genetic and pharmacological impairment to oxytocin receptor function displayed decreased aversion and increased choice toward parasitized male conspecific odour compared to control females in a mate-choice paradigm [175]. Additionally in female mice, the preference for parasitized and non-parasitized conspecific male odours was affected by the administration of an oxytocin receptor antagonist, suggesting that oxytocin is involved in the rapid decision-making associated with the approach or avoidance of a parasitic threat [48]. These findings suggest that oxytocin may play a central role in the recognition and subsequent avoidance of parasitized males by females. 

A study on conditioned disgust responses in male rats found that oxytocin modulates the expression of socially mediated disgust. When male rats with lithium chloride (LiCl)-induced sickness were conditioned with a social partner, they showed an increased mouth-gaping response (indicative of disgust and malaise) when re-exposed to the conditioned partner in comparison to exposure to a novel conspecific. Boulet et al. (2016) [176] demonstrated that when given an oxytocin receptor antagonist, rats that were conditioned with LiCl showed significantly less disgust (mouth gaping) when re-exposed to the conditioned partner, suggesting that functional oxytocin receptors are required to develop an associative disgust response toward a conspecific. As such, oxytocin may modulate both disgust-associated social salience and social recognition, integral components of mate choice.

What oxytocin-associated pathways are involved in mediating these effects remains to be determined. Also, whether these modulatory roles arise from the effects of oxytocin on other neurotransmitter/neuromodulatory pathways remains to be determined. In this regard, oxytocin has interactions with estrogens and dopamine, and likely immune components, in the regulation of social behaviours [177]. Both dopaminergic and estrogenic systems are implicated in the regulation of female mate choice and, as such, pathogen avoidance.

#### 4.2.2. Vasopressin

The neuropeptide vasopressin has also been shown to influence the avoidance of pathogen risk. Arakawa et al. (2010) [104] demonstrated that vasopressin within the MeA of rats is involved in conspecific infection avoidance. When exposed to the odours of LPS-treated male conspecifics, the expression of vasopressin messenger RNA in the MeA increased. Infusion of the vasopressin receptor (V1A) antagonist to the MeA inhibited the previously demonstrated avoidance response to LPS-treated odours. 

Vasopressin is structurally very similar to oxytocin and has a high homology for oxytocin receptors as well as vasopressin (AVP) receptors. Due to this homology, the vasopressin system may likely interact with, and influence, the oxytocin system and the role that it is known to play in pathogen and sickness detection and avoidance. Similarly, as for oxytocin, the roles of other neurotransmitter/neuromodulatory systems remain to be examined regarding the potential interactions with vasopressin.

#### 4.2.3. Estrogens

Estrogens are a class of steroid hormones that affect multiple physiological functions and behaviours in both males and females. In females, heightened sexual motivation is mainly influenced by estrogens, which in turn influence dopamine signalling in the mesolimbic reward centres of the brain [59,178,179]. The salience of a potential mate’s cues can influence the sexual preference and decision made. Lynch and Ryan (2020) [59] postulated that estrogens increase female interest in mating, while the estrogenic influence on dopamine facilitates the detection of signals with high incentive versus low incentive and informs mate preference. 

There is also evidence that estrogenic mechanisms are involved in the processing and avoidance of conspecific infection cues [175,180,181]. Kavaliers et al. (2004) [175] showed that female ERα and ERβ receptor knock-out mice had impairments in the discrimination and avoidance of urine odour in both males that were either treated with LPS or infected with a nematode parasite. This suggests that estrogenic mechanisms are involved in the recognition of, and response to, sick and infected conspecifics. Male ERα and ERβ receptor knock-out mice showed similar impairment in their response to the urine odours of sick and infected males [71]. The inability of ERα and ERβ receptor knock-out mice to recognize cues of infection was shown to not be the result of impaired olfactory processing, as other olfactory-based paradigms were not impaired (such as the ability to distinguish the sexual status of conspecifics and aversion to predator odour). 

Additionally, all three estrogen receptors (ERα, ERβ, GPER) are expressed in the MeA, where they are involved in enhancing social recognition and interact with oxytocin receptors, suggesting that estrogens are implicated in the MeA’s role in the detection of and behavioural response to parasitized conspecifics [180,182,183].

Given the roles that estrogenic mechanisms play in olfactory-mediated social recognition, response to pathogenic threat, influence on immune function, and sexual motivation/behaviour, estrogens are integral to the processing of salient social information that informs the adaptive behavioural response to pathogenic risk [180,184,185,186,187]. More research investigating how estrogenic mechanisms interact and potentially link these physiological, cognitive, and behavioural processes is required. 

#### 4.2.4. Progesterone

In human research, the compensatory prophylaxis hypothesis states that the luteal phase of the menstrual cycle (characterized by high progesterone levels) suppresses various mechanisms of immune response and thus should result in a heightened disgust response (reviewed in Fleischman and Fessler, 2011 [188]). Recent evidence suggests that progesterone plays a part in the rodent disgust response and pathogen avoidance. In a re-analysis of the results of Kavaliers et al. (2021) [189] conducted by Bressan and Kramer (2022) [190], estrous female mice administered exogenous progesterone had an enhanced avoidance response to the odours of a nematode-infected male. 

The biological mechanisms that progesterone may be acting on to influence a disgust-like response remain to be determined. However, progesterone may influence sensory inputs involved in detecting salient social information, as progesterone was shown to potentially influence social recognition [189]. More research into the potentially enhancing effects of progesterone and its neurosteroid products on pathogen detection and disgust-like response needs to be conducted before further speculation. 

#### 4.2.5. Concluding Statement on Neuropeptides and Hormones 

We have highlighted the neuromodulatory mechanisms that have been directly implicated in the modulation of pathogen detection and avoidance as they relate to mate choice. There are a range of other regulatory systems that need consideration (e.g., immune system components, opioid systems, neurotransmitters (e.g., 5-HT, endocannabinoids) and steroid and non-steroid hormones). For example, there is evidence that avoidance of infection risk is driven by physiological responses associated with stress and anxiety. Lopes (2023) [191] speculated that after the initial detection of parasite risk, the sympathetic nervous system, adrenomedullary response and adrenocortical response are triggered to facilitate avoidance. This reinforces the need to further consider the roles of, and links between, disgust, fear, and anxiety in modulating pathogen detection and avoidance. 

## 5. Conclusions

The main objective of this review was to draw attention to numerous issues of basic importance for understanding the neurobiology of pathogen avoidance and mate choice in rodents. We considered avoidance behaviours concerning potential and actual pathogenic threats, the nature of the chemical stimuli eliciting these responses, and some of the mechanisms determining and controlling responsiveness. Due to the high fitness cost of parasitism, there is strong evolutionary pressure to develop efficient systems for parasite detection and avoidance [22]. Avoidance of parasitized conspecifics is a highly conserved evolutionary trait and is universal throughout the animal kingdom [18,36,192]. To date, the majority of studies examining the neurobiology of pathogen avoidance and its relation to mate choice have been limited to laboratory conditions and have primarily considered female responses. The roles of the environmental context (e.g., presence of predators, threatening conspecifics), trade-offs with other factors (food and mate availability, own infection status), and uncertainty (what is the likelihood of the conspecific still being infectious) all need to be considered and incorporated in future studies. 

## Figures and Tables

**Figure 1 animals-14-00296-f001:**
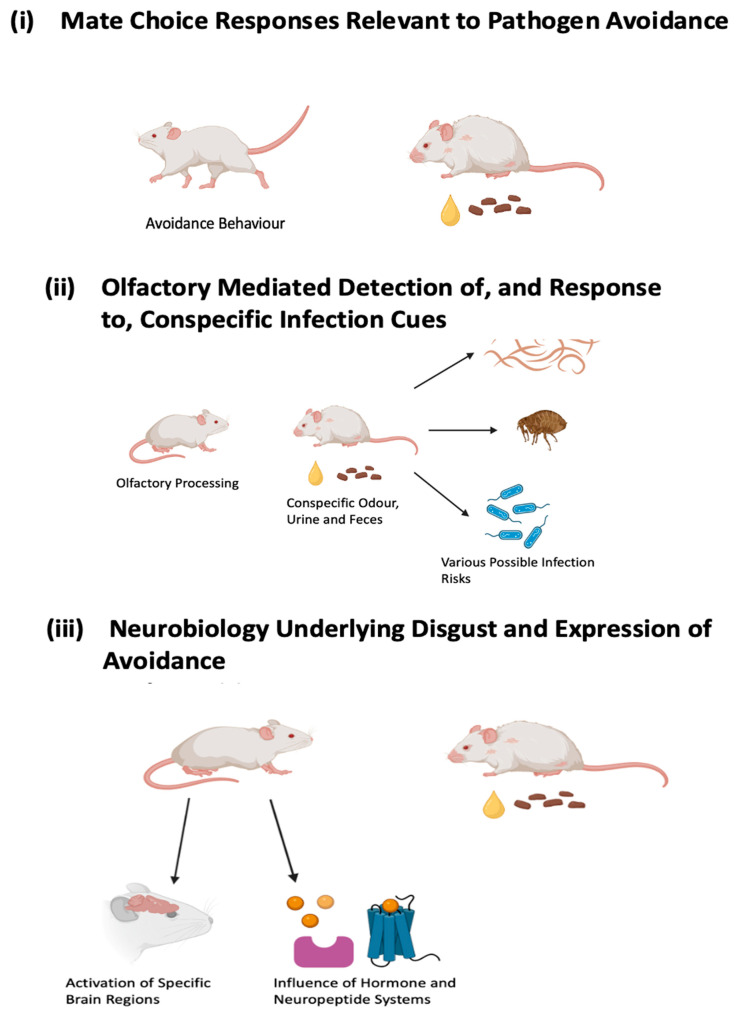
Diagram illustrating the sectional progression of the review. Section (**i**) discusses aspects of mate choice and the avoidance of pathogen threat processed in conspecific cues. Section (**ii**) discusses the detection of the pathogen risk and olfaction. Section (**iii**) discusses the neurobiological underpinnings associated with appropriate affective and cognitive responses to pathogenic cues, with a particular consideration of brain regions, hormones, and neuropeptide systems. Created with BioRender.com.

**Figure 2 animals-14-00296-f002:**
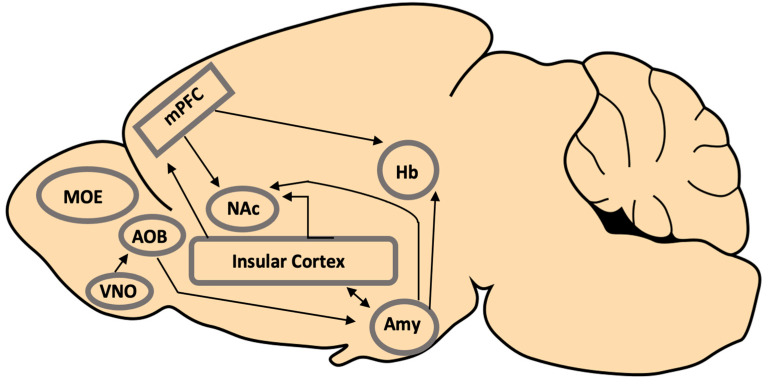
Regions and projections in the mouse brain that are proposed to be involved in the processing and mediation of the behavioural response to the pathogen risk. It is likely that additional systems involved in social information processing are also involved. MOE (main olfactory epithelium); VNO (vomeronasal organ); AOB (accessory olfactory bulb); mPFC (medial prefrontal cortex); NAc (nucleus accumbens); Hb (habenula); Amy (amygdala).

## Data Availability

The research and data discussed in this review are cited and are available in peer-reviewed journals.

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
