# Peer review of "Neurobiology of Pathogen Avoidance and Mate Choice: Current and Future Directions"

_animals, 2024, doi:10.3390/ani14020296_

Round 1

Reviewer 1 Report

Comments and Suggestions for Authors

It is an interesting and timely review of the neurobiology of pathogen avoidance and mate choice. The authors discuss different topics, including: 1) mate choice relevant to pathogen avoidance; 2) The olfactory detection of and response to conspecific cues of parasitic infection and immune status; 3) The neurobiology underlying the elicitation of disgust and expression of avoidance and aversive responses to pathogen threat. I only have some suggestions.

1.- For the less informed reader, the authors could elaborate more on the similarities and differences between mate preference and mate choice. Is it the case that the only difference is that sometimes, in nature, not all individuals have the choice to mate with the preferred individual? Maybe a table comparing both could help.

2.-The second part of the review regarding the brain regions and the hormones, neuropeptides and neurotransmitters could be better organized and described. As it stands now, they are presented as a series of evidence for each brain region. Figure 2A doesn’t add much to what is already in the text. The authors could present a tentative model of how the brain regions are activated and in which sequence to modulate the different behavioral aspects. The same can be done for the hormones, neuropeptides, and neurotransmitters.

Author Response

We have modified the manuscript according to the reviewers' comments. Specific changes that have been made are described below. We thank the reviewers for their interest and valuable comments.

1.- For the less informed reader, the authors could elaborate more on the similarities and differences between mate preference and mate choice. Is it the case that the only difference is that sometimes, in nature, not all individuals have the choice to mate with the preferred individual? Maybe a table comparing both could help.

Response:

Additional explanation of the difference between mate choice and mate preference was added. Additional citations are also included. Explicit in-text citations for detailed reviews covering mate choice/preference experiments were added (Dougherty 2020 & 2023 (citations 188 & 189)). These reviews include detailed tables breaking down relevant information. [Line 133-141]

2.-The second part of the review regarding the brain regions and the hormones, neuropeptides and neurotransmitters could be better organized and described. As it stands now, they are presented as a series of evidence for each brain region. Figure 2A doesn’t add much to what is already in the text. The authors could present a tentative model of how the brain regions are activated and in which sequence to modulate the different behavioral aspects. The same can be done for the hormones, neuropeptides, and neurotransmitters.

Response:

Figure 2A was scraped and replaced with a diagram of the mouse brain that includes the regions of interests and depicts the projections of these regions that have been found or suggested to be implicated in pathogen avoidance [Line 310-320]. 

Reviewer 2 Report

Comments and Suggestions for Authors

I appreciate the authors' efforts to write a thorough review on the topic of pathogen avoidance as it relates to mate choice. The authors provide many study examples throughout the manuscript to support their arguments, while also calling out areas where further research can and should be conducted. The manuscript is well-written and well-organized and will likely interest many readers. I have only a few minor comments for the authors to address:

I'm surprised this review did not include specific mention of the gut microbiome and its involvement in immune responses, odor production, social behavior, etc. Consider including this information within the manuscript, as there is much research on the topic and it closely relates to the content within the manuscript. Some research that may be relevant include:

https://www.sciencedirect.com/science/article/pii/S2352154615001060 

https://www.frontiersin.org/articles/10.3389/fmicb.2022.916766/full

https://www.nature.com/articles/s41396-021-00949-3

https://journals.asm.org/doi/full/10.1128/mbio.01785-15

Ln 156: Please provide citations regarding mate choice copying. There are some reviews on the topic, which could be cited here.

Ln 159: Which notable exceptions are you referring to? Please provide citations. Also, please provide citations that support the claim that animals generally demonstrate preference and choice for parasite-free mates.

Figure A2: It may be useful for readers if you edit Figure A2 into a table, then for each brain region and hormone/neuropeptide mentioned, list citations that support the claim that it is involved in rodents' behavioral response to social pathogen infection risk.

Section 4.1: There is no mention of the mesolimbic reward system within this section. Please add additional content that supports its inclusion within this section or remove the mesolimbic reward system from Figure A2.

Author Response

We have modified the manuscript according to the reviewers' comments. Specific changes that have been made are described below. We thank the reviewers for their interest and valuable comments.

1- I'm surprised this review did not include specific mention of the gut microbiome and its involvement in immune responses, odor production, social behavior, etc. Consider including this information within the manuscript, as there is much research on the topic and it closely relates to the content within the manuscript. Some research that may be relevant include:

Response: A Significant Section on the relevance of the microbiome was added to the review (Line 477-529). This section includes the recommended references plus additional citations. I believe that this section substantially adds to the review and would like to extend a thank you for suggesting its inclusion. 

2-Ln 156: Please provide citations regarding mate choice copying. There are some reviews on the topic, which could be cited here.

Response:

Two citations added to address mate choice copying (citation 152 & 153) [now line 166]

3-Ln 159: Which notable exceptions are you referring to? Please provide citations. Also, please provide citations that support the claim that animals generally demonstrate preference and choice for parasite-free mates.

Response: 

We have added citations 151, 154, 155 to address the usual preference for non-parasitized mates, and citations 156, 157  were added to address studies that are exceptions to the preferred mate being the on chosen. [Now line 173]

4-Figure A2: It may be useful for readers if you edit Figure A2 into a table, then for each brain region and hormone/neuropeptide mentioned, list citations that support the claim that it is involved in rodents' behavioral response to social pathogen infection risk

Response: 

We have changed Figure A2  into a diagram showing relevant brain regions and projections/connections that are relevant to pathogen detection/avoidance and mate choice, based on the literature that we describe in the text. [Line 310-320]

5-Section 4.1: There is no mention of the mesolimbic reward system within this section. Please add additional content that supports its inclusion within this section or remove the mesolimbic reward system from Figure A2.

Response: 

We removed mention of the Mesolimbic reward system and the original Figure A2 has been scrapped.